# Circulating Tumour DNA in Advanced Melanoma Patients Ceasing PD1 Inhibition in the Absence of Disease Progression

**DOI:** 10.3390/cancers12113486

**Published:** 2020-11-23

**Authors:** Lydia Warburton, Leslie Calapre, Michelle R. Pereira, Anna Reid, Cleo Robinson, Benhur Amanuel, Mel Ziman, Michael Millward, Elin Gray

**Affiliations:** 1School of Medical and Health Sciences, Edith Cowan University, Joondalup, WA 6027, Australia; lwarbur1@our.ecu.edu.au (L.W.); l.calapre@ecu.edu.au (L.C.); michelle.r.pereira923@gmail.com (M.R.P.); anna.reid@ecu.edu.au (A.R.); BenHur.Amanuel@health.wa.gov.au (B.A.); m.ziman@ecu.edu.au (M.Z.); 2Department of Medical Oncology, Sir Charles Gairdner Hospital, Nedlands, WA 6009, Australia; 3Anatomical Pathology, PathWest, QEII Medical Centre, Nedlands, WA 6009, Australia; Cleo.Robinson@health.wa.gov.au; 4School of Biomedical Science, University of Western Australia, Crawley, WA 6009, Australia; 5School of Medicine and Pharmacology, The University of Western Australia, Crawley, WA 6009, Australia; michael.millward@uwa.edu.au

**Keywords:** melanoma, immune checkpoint inhibitors, anti-PD1, complete response, therapy cessation, ctDNA, biomarkers

## Abstract

**Simple Summary:**

Immunotherapy is an effective treatment that harnesses the immune system to fight cancer. Drugs such as immune checkpoint inhibitors have demonstrated efficacy in the treatment of advanced melanoma. However, the optimal duration of treatment is not well established. The aim of our retrospective study was to analyse the outcomes of patients who have stopped immunotherapy treatment for advanced melanoma after durable disease control. Furthermore, we assessed circulating tumour DNA (ctDNA), which is shed from the tumour into the bloodstream, to determine its validity as a predictive biomarker of disease progression after treatment was stopped. We demonstrated that stopping treatment after durable disease control results in excellent short- to medium-term prognosis and ctDNA present at the time of stopping treatment is a strong predictor of disease recurrence.

**Abstract:**

Immunotherapy is an important and established treatment option for patients with advanced melanoma. Initial anti-PD1 trials arbitrarily defined a two-year treatment duration, but a shorter treatment duration may be appropriate. In this study, we retrospectively assessed 70 patients who stopped anti-PD1 therapy in the absence of progressive disease (PD) to determine clinical outcomes. In our cohort, the median time on treatment was 11.8 months. Complete response was attained at time of anti-PD1 discontinuation in 61 (87%). After a median follow up of 34.2 months (range: 2–70.8) post discontinuation, 81% remained disease free. Using ddPCR, we determine the utility of circulating tumour DNA (ctDNA) to predict progressive disease after cessation (*n* = 38). There was a significant association between presence of ctDNA at cessation and disease progression (*p* = 0.012, Fisher’s exact test) and this conferred a negative and positive predictive value of 0.82 (95% CI: 0.645–0.930) and 0.80 (95% CI 0.284–0.995), respectively. Additionally, dichotomised treatment-free survival in patients with or without ctDNA at cessation was significantly longer in the latter group (*p* < 0.001, HR: 0.008, 95% CI: 0.001–0.079). Overall, our study confirms that durable disease control can be achieved with cessation of therapy in the absence of disease progression and undetectable ctDNA at cessation was associated with longer treatment-free survival.

## 1. Introduction

The treatment options for metastatic melanoma have improved dramatically with the advent of novel therapies including checkpoint inhibitors, leading to substantial improvement in progression-free survival (PFS) and overall survival (OS), heralding a new era in melanoma treatment [1,2]. Current first-line immunotherapy options in advanced melanoma include either anti-PD1 (anti programmed cell death receptor 1) antibody monotherapy or the combination of ipilimumab (anti CTLA4 antibody—Cytotoxic T lymphocyte-associated protein 4) and nivolumab (anti-PD1). The immune checkpoint inhibitor pembrolizumab received FDA approval in 2014 for patients with advanced melanoma [3]. Pembrolizumab is an engineered humanised IgG4 monoclonal antibody that regulates T cell activation by blocking the programmed cell death protein 1 (PD1) [4]. 

Unfortunately, only a minority of patients experience the dramatic benefits of immunotherapy due to primary and acquired resistance [5]. Complete response (CR) occurs in up to 20% of metastatic melanoma patients treated with anti-PD1 monotherapy [2]. The duration of immunotherapy treatment in responding patients has been arbitrarily set at 2 years but optimal duration remains unknown [6,7]. Although anti-PD1 monotherapy is a relatively well-tolerated treatment, there remains a risk of toxicity, and little is known about the long-term effects. In addition, ongoing therapy impacts patient quality of life and health care budgets. Dealing with these burgeoning costs is a major challenge for the health system and considerable research is being conducted to find biomarkers to identify responders to appropriately tailor treatment choices. In the meantime, identifying patients that can safely undergo a shorter treatment duration and save health care dollars is paramount.

Circulating tumour DNA (ctDNA), released into plasma from apoptosing or necrotic tumour cells, has emerged as a useful biomarker for assessment of response to therapy in various cancers [8,9,10]. In melanoma, we and others have shown the clinical validity of using ctDNA for tracking response to therapy and prognostication [11,12,13,14,15]. Baseline ctDNA levels have been shown to be directly associated with radiological tumour burden and inversely associated with response and PFS [16]. Furthermore, it has been shown that undetectable ctDNA at baseline or within eight weeks of commencing anti-PD1 immunotherapy is an independent predictor of response and PFS [12,15]. Nonetheless, there have been no studies to date to investigate ctDNA as an indicator for safe early cessation of therapy and durable treatment-free survival (TFS). Such a test would be especially useful in difficult or equivocal radiological evaluations where tissue biopsy is not feasible.

Here, we retrospectively assessed and report the clinical outcomes of 70 patients who stopped anti-PD1 therapy in the absence of progressive disease (Figure 1). In addition, we explore the potential of ctDNA to determine safe cessation of therapy and detection of early recurrence. 

## 2. Results

### 2.1. Patient Description

Seventy patients were identified who ceased pembrolizumab after achieving durable disease control (stable disease (SD), partial response (PR) or complete response (CR)) without evidence of progressive disease at time of treatment cessation. Baseline demographics (*n* = 70) are shown in Table 1. Median age was 71 (29–91) years. Most patients had M1c (44%) disease (AJCC eighth edition) with normal LDH levels (70%) and less than three metastatic sites of disease (77%). Half of the cohort had received prior therapy including chemotherapy (7%), ipilimumab (21%) and targeted therapy (19%). BRAF V600 mutations were present in 14 patients (20%), with 13 having received and progressed on prior BRAF inhibitor treatment. Only seven patients (10%) had intracranial disease involvement at the time of treatment initiation.

Median treatment duration was 11.8 (range: 3–33) months, with 38 patients (54%) having less than 12 months of treatment. A total of 61 out of the 70 (87%) patients attained CR. The median duration of therapy for patients who achieved CR was 11.7 months (range: 3–28), whilst the median duration of therapy for patients attaining PR/SD was 24.7 months (range: 5–33). The median follow-up since treatment cessation was 34.2 months (range: 2–70.8) (Table 2, Figure 2). Median PFS and OS for the cohort have not yet been reached (Figure 3A,B). Twelve patients (17%) ceased due to immune-related toxicity (*n* = 9) or intolerance (*n* = 3). Reasons for treatment cessation due to toxicity included Guillain–Barre Syndrome (*n* = 1), colitis (*n* = 1), hepatitis (*n* = 1), pneumonitis (*n* = 2), skin toxicity (*n* = 3) and pancreatitis (*n* = 1).

### 2.2. Progressive Disease in the Whole Cohort

Thirteen patients (19%) developed progressive disease following cessation of therapy. Disease progression occurred with the following best overall response (BOR) distribution: SD: 2/3, PR: 2/6 and CR: 9/61. The median time to disease progression following cessation of therapy for all patients was 12.1 months (range: 2.2–27.2). The median time on therapy for patients with CR who subsequently recurred was similar to the whole cohort at 11.7 months. With respect to the baseline characteristics of patients with PD, 12/13 had less than three metastatic sites of disease, 8 had M1c disease, none had brain metastases prior to treatment initiation and two patients had an ECOG greater than 1.

### 2.3. ctDNA in Patients Ceasing Anti-PD1 Therapy

We analysed ctDNA levels at the time of immunotherapy cessation in the subgroup of patients with assessable blood samples (Figure 1). A total of 38 patients included in the study had a traceable mutation and a blood sample collected within 16 weeks prior to or after anti-PD1 immunotherapy cessation (Appendix A). A total of 10/38 patients developed disease progression whilst the rest of the cohort (28/38) sustained response post immunotherapy cessation. The mutations identified in the patients’ tumours and used for ctDNA assessment are listed in Appendix A. Five of the 38 patients had detectable ctDNA at the time of treatment cessation but only 4 of these patients went on to develop progressive disease within 2.5–14.2 months (median: 8.4 months) following cessation (BOR; CR = 2/4, PR = 2/4). Two of the five patients with detectable ctDNA (MM176 and MM295) had rising ctDNA levels prior to cessation, suggesting treatment resistance despite ongoing radiological response at cessation (Appendix A). The other six patients with radiological progression had CR as their best response, undetectable ctDNA at cessation and developed disease progression within 4.7–21.8 months (median: 15 months). One patient with detectable ctDNA at cessation did not develop disease progression at time of analysis.

Of the 28 of 38 patients analysed for ctDNA who did not progress, 26 patients achieved CR, 1 had PR and 1 SD at cessation of therapy. Plasma ctDNA was undetectable in 27 of these 28 (96%) patients, with only 1 patient having detectable ctDNA at cessation (Table 3). CtDNA became undetectable 83 weeks after cessation and the patient remained in CR until last follow up at 45 months after cessation (Appendix A).

As shown in Table 3, the rate of detectable ctDNA without disease progression or false-positive results was low, with a specificity of 0.96. However, the rate of undetectable ctDNA in patients who developed disease progression or false negatives was much higher, resulting in a poor sensitivity of 0.33.

Detectable ctDNA at cessation conferred a high relative risk of relapse (RR: 4, 95% CI: 1.88–10.26) and a positive predictive value of 0.80 (95% CI 0.283–0.995). By contrast, undetectable ctDNA at cessation was significantly associated with a low risk of relapse by Fisher’s exact test (*p* = 0.012, Table 3), with a negative predictive value of 0.82 (95% CI: 0.645–0.930). Additionally, absence of ctDNA at cessation was significantly associated with longer treatment-free survival (*p* < 0.0001, HR: 0.007, CI 95% (0.001–0.786). The median treatment-free interval for patients with detectable ctDNA at time of treatment cessation was 10 months but was not reached in patients with undetectable ctDNA (Figure 3C).

## 3. Discussion

In this study, we describe a real-world local experience and outcomes of melanoma patients who stopped anti-PD1 therapy in the absence of disease progression, contributing to the growing body of research demonstrating durable response to immunotherapy [17,18,19]. Furthermore, we have a subgroup of these patients to determine the utility of ctDNA in predicting recurrence following cessation. Our study is the first to evaluate ctDNA in the context of treatment cessation in advanced melanoma.

The arbitrary treatment duration of immunotherapy for advanced melanoma patients imposes clinical and financial toxicity on patients. There is increasing interest in determining the optimal treatment duration for immunotherapy in advanced melanoma. Several clinical trials have commenced recruitment to address this question (STOP GAP NL65512.078.18, Safe Stop NTR7502). Two pivotal studies, Checkmate 067 and Keynote 006 evaluated different lengths of treatment duration; Keynote 006 mandated a maximum of 2 years of treatment, whilst Checkmate 067 did not define a maximum treatment period [20,21]. In a four-year survival update for Checkmate 067, one-quarter of patients who received anti-PD1 monotherapy still remained on treatment [22]. However, at five years, this dropped to 7% without any significant change in overall survival [21]. 

With over 50% of patients in our cohort having less than 12 months of treatment and the recurrence rate remaining comparable to Keynote 006, it raises the possibility of safe early treatment cessation once CR has been confirmed. Our cohort of 70 patients certainly further confirms that beyond clinical trials, patients who attain complete response to anti-PD1 monotherapy have an excellent prognosis in the short to medium term and this gives further credence to the concept of stopping treatment early following CR [7].

With respect to the whole cohort, our data are comparable to other immunotherapy cessation reports [17,19,23,24]. The median time on treatment for our cohort was 11.8 months (range: 3–33), with a similar median treatment time for patients who achieved a complete response (11.7 months). This is similar to the recently published largest real-world cohort of 187 patients who stopped treatment in the absence of disease progression with a median time on therapy of 12 months [17]. In our study 38 patients (54%) had less than 12 months of treatment. In those who had disease progression, the median time on therapy was similar to the whole cohort at 12.4 months. However, 33% had 6 months or less of treatment. This is concordant with other studies showing treatment less than 6 months has been shown to be associated with a higher risk of progression [17]. 

With a median of 34 months follow up after treatment cessation, progression occurred in 19% of our cohort. This recurrence rate is comparable to other studies which report progressive disease in 11–23% of patients stopping treatment [17,19,23,24]. It appears that if 6–24 months of therapy is completed, the risk of progression is similar, regardless of the duration of therapy [17]. Based on clinical data from multiple treatment cessation studies, albeit post hoc or retrospective, it is becoming clearer that the risk of recurrence following CR is in the order of 13–24% and the rate of recurrence does not appear to be affected by treatment duration [2,17,19].

Undetectable ctDNA prior to and/or after cessation appears to strongly predict lack of recurrence and detectable ctDNA moderately predicts relapse in these patients. Thus, ctDNA in this population can become a useful adjunct to monitoring with standard imaging criteria. Furthermore, a detectable ctDNA level following treatment cessation predicts disease progression and shorter treatment-free intervals, and should, therefore, prompt radiological surveillance and close observation.

Patients with PR or SD as their best response have a higher risk (14–32%) of developing disease progression following treatment cessation [2,17,24]. The rate of progression is also high in our SD/PR (4/9, 44%) vs. CR (9/61, 15%) group but the low number of the former is not sufficient to make definitive conclusions about the risk of disease progression in this group. With the higher rates of recurrence in patients with PR/SD, it is clear that guidelines for the duration of treatment are particularly needed in this population. We have demonstrated that serial monitoring of ctDNA around the time of cessation in this population may provide valuable information about the disease kinetics, assisting in treatment decisions. For example, rising ctDNA levels despite ongoing radiological response could guide early treatment cessation and switch. Alternatively, ctDNA may be used to guide treatment length in PR/SD patients with declining or undetectable ctDNA levels. For these patients, therapy could be stopped with the potential inclusion of a biopsy of any residual radiologic disease for confirmation of pathological response. Given the potential pharmacoeconomic benefits of this approach, we feel prospective trials on treatment cessation should incorporate plasma ctDNA testing. This will be useful to validate ctDNA as an added biomarker for guiding treatment cessation in patients with partial response or stable disease.

While our study does provide some evidence of the potential of ctDNA as a disease status biomarker, we acknowledge sampling error as a significant caveat to the use of ctDNA in monitoring post therapy cessation in patients with CR. Given the low disease burden in the cessation cohort, the rate of false-negative results in those with undetectable levels was high. While our use of digital PCR provides a high technical sensitivity for analysis of ctDNA, this targeted approach is highly sensitive to sampling error at such low ctDNA concentrations [10,25]. In the analysis of ctDNA in our cohort, the specificity to detect relapse was high, but the sensitivity was disappointingly low. Sensitivity may be enhanced with the use of larger plasma volumes and the monitoring of multiple mutations for each case [26]. It has been demonstrated that ctDNA can certainly be valuable in minimal residual disease [27] and therefore further studies to evaluate the sensitivity and specificity in this post treatment setting are warranted to progress its integration into clinical use.

## 4. Materials and Methods

### 4.1. Patients

Seventy patients treated for stage IV melanoma with intravenous anti-PD1 monotherapy (Pembrolizumab, 2mg/kg every three weeks) at Sir Charles Gairdner Hospital in Perth, Western Australia between August 2013 and July 2019 were retrospectively evaluated (Figure 1). Patients were included if they ceased anti-PD1 monotherapy due to CR or had treatment for two years and stopped with a durable response (SD or PR). Patients were also included if they ceased treatment due to toxicity but had no evidence of disease progression at cessation of therapy. There was no defined minimum duration of treatment included. Written informed consent was obtained from all patients under approved Human Research Ethics Committee protocols from Edith Cowan University (No. 11543 (9 July 2014) and No. 18957 (14 May 2018)) and Sir Charles Gairdner Hospital (No. 2007-123 (1 October 2007), No. 2013-246 (5 June 2014), RGS0000003289 (24 June 2020)) in compliance with the Declaration of Helsinki. Experiments were performed in accordance with institutional and national guidelines and regulations.

Demographic and clinical data including age, gender, Eastern Cooperative Oncology Group (ECOG) performance status, disease stage, baseline LDH and presence of brain metastases were collected. The duration of pembrolizumab therapy, time to BOR, recurrence and sites of progression were assessed. Subsequent treatments and response were also collected. Tumour responses during and at completion of treatment were assessed radiologically by computer tomography (CT) and/or positron emission tomography (PET) scans by Response Evaluation in Solid Tumors (RECIST 1.1) or immune-related Response Criteria (irRC) and classified as having a complete response, partial response or stable disease. CR was defined as an absence of radiological disease or a negative biopsy of residual tissue.

### 4.2. Blood Collection and ctDNA Analysis

Blood samples were collected using a EDTA vacutainer or cell-free DNA BCT tubes (Streck, La Vista, NE, USA). Plasma was separated from whole blood as previously described and stored at 4 °C [27,28]. Cell-free DNA (cfDNA) was extracted from 5 mL of plasma using the QIAamp Circulating Nucleic Acid Kit (Qiagen, Hilden, Germany). DNA samples were stored (−80 °C) until analysis. The ctDNA was quantified by droplet digital PCR (ddPCR).

Trackable mutations were identified in 45/49 (92%) patients with samples available via standard pathology protocols or using a customised melanoma NGS panel (Illumina, San Diego, CA, USA), as described by Calapre et al. [29]. Commercially available and/or customised probes were used to analyse ctDNA by ddPCR [30]. Amplifications were carried out in a 20 μL reaction containing 1× droplet PCR supermix, 250 nM of each probe, 900 nM primers and 5 or 8 μL cfDNA. Droplets were generated and analysed using the QX200 system (Bio-Rad, Hercules, CA, USA). The threshold for each assay used for ctDNA detection was previously reported by Calapre et al. [29] and Marsavela et al. [31]. A positive control, a healthy control and a no template control were included in each run. Only tests providing more than 10,000 droplets were used for analysis. Quantification results were presented in copies of ctDNA per mL of plasma. All samples were tested in triplicate. 

### 4.3. Statistics

Median OS, PFS and TFS were calculated using the Kaplan–Meier method and compared using the log-rank test. Correlation between ctDNA detectability at cessation and disease progression, with corresponding *p*-values, positive predictive values (PPV) and negative predictive values (NPV) were calculated using Fisher’s exact test. All statistical analyses were performed using GraphPad Prism version 8 (GraphPad Software Inc., San Diego, CA, USA) and SPSS version 25 (IBM, Armonk, NY, USA).

## 5. Conclusions

Our study provides additional evidence that cessation of treatment in the context of complete response is durable. Given the economic toxicity, side effects and impact on quality of life of ongoing treatment, it is important to determine optimal treatment duration and establish biomarkers to guide treatment cessation decisions. Plasma ctDNA at cessation appears to predict recurrence. However, its clinical utility is still limited by its capacity to reliably detect residual disease. As pre-analytical conditions and methodology improve, we anticipate that ctDNA could ultimately be an important adjunct to treatment cessation decisions.

## Figures and Tables

**Figure 1 cancers-12-03486-f001:**
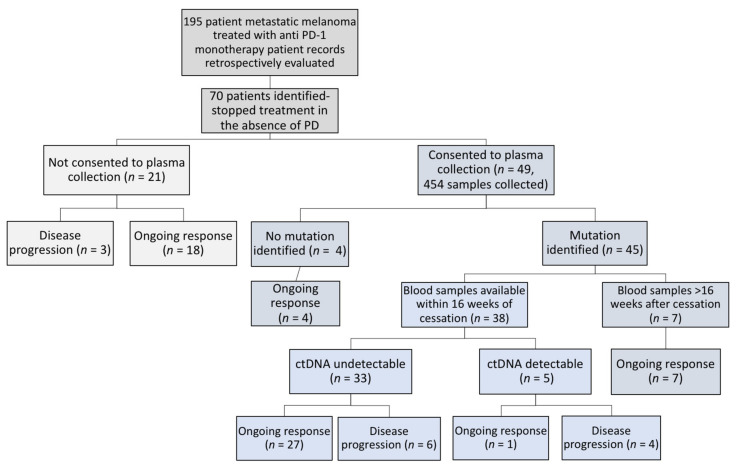
Schema of the 70 patients treated with anti-PD1 therapy and ceased treatment in the absence of disease progression.

**Figure 2 cancers-12-03486-f002:**
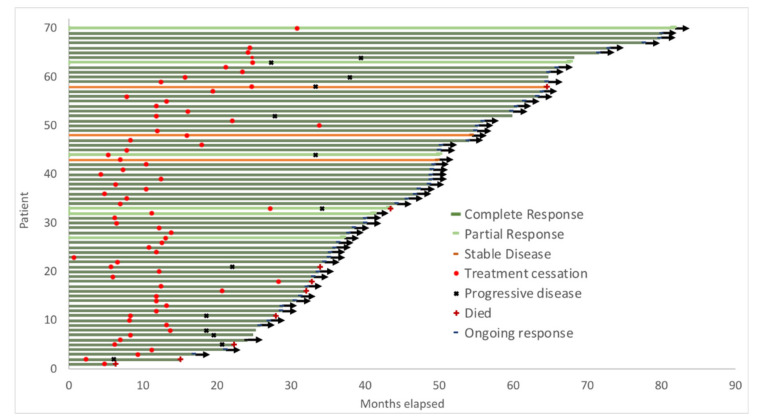
Swimmer plot indicating time on anti-PD1 therapy, progression-free survival and overall survival following disease progression.

**Figure 3 cancers-12-03486-f003:**
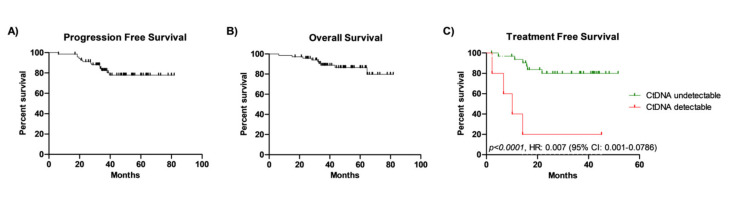
Survival outcomes of melanoma patients that ceased anti-PD1 immunotherapy. Kaplan–Meier probability curves for progression-free survival (PFS, **A**) and overall survival (OS, **B**) for the total cohort of patients who discontinued treatment in the absence of disease progression (median PFS and OS not yet reached. (**C**) Dichotomised treatment-free survival (TFS) of patients detectable or undetectable ctDNA at cessation (*p* < 0.001, log-rank test, median TFS not yet reached).

**Table 1 cancers-12-03486-t001:** Patient demographics of the study cohort prior to immunotherapy initiation (*n* = 70).

Demographic Variables	Number of Patients (%)
Age (years)	
Median	71
Range	29–91
Gender	
Male	49 (70)
Female	21 (30)
ECOG	
0	53 (75)
1	11 (16)
≥2	6 (9)
Stage at commencement of anti-PD1	
M1a	20 (29)
M1b	12 (17)
M1c	31 (44)
M1d	7 (10)
Number of metastatic sites	
<3	54 (77)
>3	16 (23)
Lines of prior therapy	
0	36 (51)
≥1	34 (49)
Prior systemic therapy	
Chemotherapy (DTIC/Fotemustine)	5 (7)
BRAFi or BRAF/MEKi	13 (19)
Ipilimumab	15 (21)
BRAF V600 mutation	
Yes	14 (20)
No	56 (80)
LDH prior to anti-PD1 therapy	
<ULN	49 (70)
>1×ULN	13 (19)
>2×ULN	3 (4)
Unknown	5 (7)
Brain metastases	
Absent	63 (90)
Present	7 (10)

Abbreviations: ECOG, Eastern Cooperative Oncology Group; AJCC, American Joint Committee on Cancer (8th edition); DTIC, Dacarbazine; LDH, lactate dehydrogenase; ULN, upper limit of normal.

**Table 2 cancers-12-03486-t002:** Outcome of anti-PD1 antibody treatment (*n* = 70).

Variable	*N* (%)
Treatment duration (months)	
Median	11.8
Range	3–33
Response at cessation	
Complete Response	61 (87)
Partial Response	6 (8.6)
Stable Disease	3 (4.3)
Time to BOR (months)	
Median	5
Range	1–29
Time to CR (*n*= 61) (months)	
Median	5
Range	1–29
PD post cessation	
No	57 (81)
Yes	13 (19)
Follow up post cessation (months)	
Median	34.2
Range	2–70.8
Treatment free survival (months)	
Median	Not reached
Range	2–70.2
Time to PD after cessation (n=13) (months)	
Median	11.1
Range	2.2–27.8
PFS (months)	
Median	Not reached
Range	6–81.8
OS (months)	
Median	Not reached
Range	6.3–81.8

**Table 3 cancers-12-03486-t003:** Plasma ctDNA results at time of cessation relative to clinical outcome.

	Ongoing Response (*n*)	Disease Progression (*n*)	Total (*n*)
ctDNA detectable	1	4	5
ctDNA undetectable	27	6	33
Total	28	10	38

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
