# Peer review of "Circulating Tumour DNA in Advanced Melanoma Patients Ceasing PD1 Inhibition in the Absence of Disease Progression"

_cancers, 2020, doi:10.3390/cancers12113486_

Round 1

Reviewer 1 Report

This is a very interesting topic but th presentation of results isn't very clear

In the flow chart: 195 patients -->70 stopped ICI and only 49 participated. So why in Table 1--> 100 patients described in this table. Why 100? wht do they correspond to?

Table1-->"brain metastases": prior to ICI initiation?

It would be interesting to have data concerning the duration of tretament and outcome at treeatment cessation-->the patients with SD/PR at treatment interruption : were they treated longer than those who achieved CR?

"With respect to the patients with PD, 12/13 had less than three metastatic sites of disease, 8 had M1c disease, none had brain  metastases prior to treatment initiation and two patients had an ECOG greater than 1. " -->All these caracteristics are Baseline of ICI initiation or at progression?

"4 of these 38 patients had detectable ctDNA"--> but there are 5 in table 3? why?

Could you precise the status of these 4 (or 5?) patients at ICI cessation: PD?CR? SD?

Among the 38 patients none had several blood samples? it was only one dosage,  either at <16 or >16 weeks?

Author Response

Thank you for taking the time to review our manuscript and the suggestions provided to further improve it. Please see below the individual responses to each of the points raised. Changes are tracked within the manuscript.

  1. This is a very interesting topic but the presentation of results isn't very clear.

The results section has been re-written to make it clearer and easier to interpret alongside the tables and graphics. We have also included some more data in the supplementary section – table S2.

  1. In the flow chart: 195 patients -->70 stopped ICI and only 49 participated. So why in Table 1--> 100 patients described in this table. Why 100? What do they correspond to?

In table 1 – there are 70 patients included (ie 49 males and 21 females) – the percentages that these represent are listed in table 1 within brackets, next to the raw number. This number adds up to 100 and may be what you are referring to? This is a percentage of the total cohort. We have indicated ‘(n=70)’ in the figure heading to clarify this.

  1. Table1-->"brain metastases": prior to ICI initiation?

Yes. This represents brain metastasis prior to ICI initiation. This was alaredy included in the results section line 95: “Only seven patients (10%) had intracranial disease involvement at the time of treatment initiation”. This has also now been added to the table title for clarity.

  1. It would be interesting to have data concerning the duration of treatment and outcome at treatment cessation-->the patients with SD/PR at treatment interruption : were they treated longer than those who achieved CR?

The referenced immunotherapy trials including Keynote 001 and 006 had protocols whereby patients who had a minimum of 6 months of therapy and attained complete response were eligible to stop treatment. This trial also defined a maximum treatment of two years. Due to the published increased chance of durable control after stopping treatment in the presence of complete response, it is usually only these patients who are offered cessation prior to 2 years in the real-world setting. Therefore, it is not unexpected that those who achieve SD or PR do have longer treatment durations than those who achieve CR. And it is therefore not the length of treatment that is predictive of durable response but the depth of response. This was commented on in the discussion. However, we have now added the duration of treatment for CR and PR/SD group into the results section in line 104-105:

“…, whilst the median duration of therapy for patients attaining PR/SD was 24.7 months (range: 5-33).”

The outcome of treatment cessation in SD/PR patients was already included for the whole cohort (n=70), in line 127-128: “Disease progression occurred with the following best overall response (BOR) distribution; SD: 2/3, PR: 2/6 and CR: 9/61”.

Furthermore, the outcomes of patients with SD/PR in the cohort with ctDNA available has been graphically represented in the supplementary material (Figure S1A-B). We hope this adequately addresses this comment.

  1. "With respect to the patients with PD, 12/13 had less than three metastatic sites of disease, 8 had M1c disease, none had brain  metastases prior to treatment initiation and two patients had an ECOG greater than 1. " -->All these characteristics are Baseline of ICI initiation or at progression?

Yes, this represents baseline characteristics. The text in line 131-132 has been altered to clarify this.

  1. "4 of these 38 patients had detectable ctDNA"-->but there are 5 in table 3? why?

“4 of these 38 patients had detectable ctDNA at the time of treatment cessation and these patients experienced progressive disease within 2.5-14.2 months following cessation.”

There were 5 patients with detectable ctDNA at baseline. Only 4/5 of these patients had disease progression. That is, 4/38 patients had detectable ctDNA and disease progression. 1/38 patients had detectable ctDNA and no disease progression. This is what is represented in Table 3 and in Figure 1. The results section has been significantly expanded to better explain this (Lines 139- 158).

  1. Could you precise the status of these 4 (or 5?) patients at ICI cessation: PD?CR? SD?

The break down on best response was originally included in the manuscript for the whole cohort (n=70) in line 127-128. We have now amended the results section to include the breakdown for the 4/38 patients with PD and ctDNA available in line 142-144:

“Five of the 38 patients had detectable ctDNA at the time of treatment cessation but only 4 of these patients went on to develop progressive disease within 2.5-14.2 months (median: 8.4 months) following cessation (BOR; CR=2/4, PR=2/4).”

  1. Among the 38 patients none had several blood samples? it was only one dosage,  either at <16 or >16 weeks?

As this was a retrospective paper, blood collections were inconsistent. We have included further information in the supplementary section (Supplementary Table 2) which outlines the weeks before or after cessation at which patient blood collections were made, their clinical outcomes and the copies/mL of ctDNA. As you can see from this table, some patients had blood collected prior to, at or after cessation whilst other patients had samples collected at multiple time points.

Reviewer 2 Report

Warburton and colleagues showed a retrospective study on stage IV melanoma, where they evaluated the detectable levels of ctDNA to determine its correlation to disease progression.

While well described, it is disappointing to get to the end of the manuscript and realize the sampling error. For that reason, the authors should earlier on (in results) include at least the specificity and sensitivity of their analysis. Ideally, the authors should be able to increase the sample number and improve sensitivity of the method as discussed.

Regarding Introduction, while the authors only refer to immunotherapy options, they should include what is the molecular basis for those therapeutic decisions. In addition, they could also better discuss why only a minority of patients actually benefit from those treatments.

In Results, while the authors state they “analysed ctDNA levels at the time of immunotherapy cessation” (line 125), they later say those patients had “a blood sample within 16weeks prior to and post anti-PD1 cessation.” Does this means there were samples from patients while still on therapy (“prior to … cessation”), which were included with those obtained until 16weeks after cessation? Please clarify.

Since the authors have identified different mutations, it would be interesting to relate the specific mutations present in patients to ctDNA levels and outcome.

Generally, there is a lack of results detailed description, so results shown in the Tables should be fully described in the text.

Minor points

In Figure 1, which mutations were identified should me mentioned.

The terms “ctDNA negative” and “ctDNA positive” present in Figure 1 should be replaced by "undetectable" and "detectable", respectively, consistent with what is referred in the text and Figure 3.

Tables’ legends should include the abbreviations used (ECOG, LDH, M1a, DTIC, ULN, etc).

There is a discrepancy in the CR patients in text (n=64) vs Table 2 (n=61).

Please review line 136 as there’s an extra ctDNA seemly out of place.

Author Response

Thank you for taking the time to review our manuscript and the suggestions provided to further improve it. Please see below the individual responses to each of the points raised. Changes are tracked within the manuscript.

  1. While well described, it is disappointing to get to the end of the manuscript and realize the sampling error. For that reason, the authors should earlier on (in results) include at least the specificity and sensitivity of their analysis. Ideally, the authors should be able to increase the sample number and improve sensitivity of the method as discussed.

We were also disappointed in the sensitivity. We did mention this in the results in reference to the PPV and NPV, but we have now included the sensitivity and specificity in the results section and deleted the absolute values from the discussion.  Lines 155-158- read:

“As shown in Table 3, the rate of detectable ctDNA without disease progression or false positive results was low, with a specificity of 0.96. However, the rate of undetectable ctDNA in patients who developed disease progression or false negatives was much higher, resulting in a poor sensitivity of 0.33.”

However, as this is a retrospective paper it is difficult to enhance the sample size. As it is the intra-patient sample size that ultimately needs enhancing, this cannot be done on our retrospective cohort. That is, ideally each patient needs larger volume of plasma collected and with increased frequency of blood collections around the time of cessation to improve the likelihood of detecting ctDNA. This would likely reduce the numbers of false negative results.

The sensitivity is unfortunately known to be poor for ctDNA but is not often overtly reported. In a paper referenced by us by RJ Lee et al (ref 27), they looked at minimal residual disease using ctDNA in an earlier setting- post curative resection of stage III melanoma- detectable ctDNA was very good at predicting relapse as it was in our data but undetectable ctDNA did not exclude relapse, with a relatively high false negative rate. They had 52% of patients who had undetectable ctDNA after resection go on to ultimately relapse.

Similarly, JH Lee et al. (ref 28) examined the predictive role of detectable ctDNA prior to curative surgery for high risk stage II/III melanoma patients (n=119). At the time of their analysis, 96 patients (81%) patients had disease recurrence. Whilst only 40 out of the 96 who recurred (42%) had detectable ctDNA pre-operatively indicating that the absence of ctDNA does not strongly predict the absence of disease recurrence.

Improving the sensitivity of ctDNA is something our lab is working on currently, as we appreciate that its clinical application is limited unless this is addressed. We therefore made comment on the limitations of the blood sampling in our cohort to highlight the issue so it is reported more consistently and considered in future prospective ctDNA trials in terms of protocol development. We are unable to enhance the data for the cohort of patients with retrospectively collected blood samples and feel that simply adding more patients with retrospectively collected data is unlikely to enhance the sensitivity.

  1. Regarding Introduction, while the authors only refer to immunotherapy options, they should include what is the molecular basis for those therapeutic decisions. In addition, they could also better discuss why only a minority of patients actually benefit from those treatments.

Whilst we appreciate your point here, it is difficult to succinctly summarise the treatment landscape here in Australia which is bound by regulatory funding bodies that impact treatment sequencing choices. We did include the data about prior treatment including the proportion of the cohort who had targeted therapy for BRAF mutation prior to PD-1 inhibition in Table 1.

Aside from BRAF/MEK inhibitors for BRAF mutant melanoma, there are no FDA approved treatments that guide treatment decisions based on mutational changes. Currently patients either receive checkpoint inhibitors or BRAF directed targeted therapy as first line standard of care. Therefore outside of the presence or absence of BRAF v600 mutations or clinical trials, currently there is no molecular basis for therapeutic decisions.

Finally, the understanding of why only a minority of patients benefit from immunotherapy continues to evolve, as there is further retrospective analysis of genetic signatures and cellular immune responses that predict a favourable response or non-response. At five years, only 44% of patients treated with anti-PD1 monotherapy in clinical trials are alive confirming that for the majority of patients treated, durable response is unfortunately not achieved. Discussion, about the reasons for primary or acquired resistance including T cell exhaustion, genetic mutations and neoantigens is a very extensive topic and we felt that it would not be able to be discussed well within the context of this paper.

We added the following in the introduction and the link to a comprehensive review on this topic, in line 59 …due to primary and acquired resistance [5].”

  1. In Results, while the authors state they “analysed ctDNA levels at the time of immunotherapy cessation” (line 125), they later say those patients had “a blood sample within 16weeks prior to and post anti-PD1 cessation.” Does this means there were samples from patients while still on therapy (“prior to … cessation”), which were included with those obtained until 16weeks after cessation? Please clarify.

As this was a retrospective study, blood collection timepoints were inconsistent. We defined plasma available within the cessation timeframe as being within 16 weeks either side of cessation. We excluded patients who did not have blood collections within this period. Some patients had multiple blood collections either before, on the day of cessation or after cessation. We have now included a supplementary table to demonstrate the blood collection time points and integrated this information with radiological response, mutational data, ctDNA levels and treatment free survival.

There were only 4 patients who had blood collections prior to treatment cessation only (Range 3-10 weeks prior to cessation). The blood samples for all these patients were undetectable. Only one of these patients with an isolated pre treatment sample went onto develop disease progression and this patient had their sample only three weeks prior to treatment cessation.

Where multiple blood samples were available, we used the blood collection at or post cessation.

  1. Since the authors have identified different mutations, it would be interesting to relate the specific mutations present in patients to ctDNA levels and outcome.

We have now included this information in table S2. As you can see, there was no identified pattern in relation to mutation present. We have therefore not elaborated on this in our discussion.

  1. Generally, there is a lack of results detailed description, so results shown in the Tables should be fully described in the text.

Results section has been re-written and elaborated to better describe the data.

Additionally, whilst we have tried to avoid describing individual cases in the body of the text, we have included more information within the supplementary material.

Minor points

In Figure 1, which mutations were identified should me mentioned.

There were 22 different mutations identified in 45 patients- it is not possible to fit them into figure 1. They are listed separately in the supplementary material in Table S1 and S2.

The terms “ctDNA negative” and “ctDNA positive” present in Figure 1 should be replaced by "undetectable" and "detectable", respectively, consistent with what is referred in the text and Figure 3.

Thank you. This has been amended.

Tables’ legends should include the abbreviations used (ECOG, LDH, M1a, DTIC, ULN, etc).

Added as requested.

There is a discrepancy in the CR patients in text (n=64) vs Table 2 (n=61).

Apologies. Text should read n=61. This has been corrected. 

Please review line 136 as there’s an extra ctDNA seemly out of place.

Corrected. Thank you.

Reviewer 3 Report

Dear authors,

It was with great interest I read your manuscript concerning ctDNA in patients that stopped PD-1 inhibition. A very well written and concise paper on a very important subject. My minor comments are:

  1. Line 52: I would rather write “Current first line immunotherapy options..”, since there are also e.g. targeted therapies available as first-line treatments for selected patients (which 19% of the patients in this cohort received).
  2. Line 68: ctDNA can be released from cells undergoing other types of cell-death than apoptosis (and for example the length of the DNA fragments can be used to differentiate between e.g. apoptosis and necrosis). To make it even more complex, ctDNA can also be secreted. You do not have to go deep, but please revise the sentence.
  3. The numbers in Figure 1 and Table 3 are not the same, seems like one recurrence is reported differently.
  4. Results: Why not also report specificity (96%) and sensitivity (40%)?

Author Response

Thank you for taking the time to review our manuscript and the suggestions provided to further improve it. Please see below the individual responses to each of the points raised. Changes are tracked within the manuscript.

1. Line 52: I would rather write “Current first line immunotherapy options..”, since there are also e.g. targeted therapies available as first-line treatments for selected patients (which 19% of the patients in this cohort received).

Agree. This is more accurate. It has been changed.

2. Line 68: ctDNA can be released from cells undergoing other types of cell-death than apoptosis (and for example the length of the DNA fragments can be used to differentiate between e.g. apoptosis and necrosis). To make it even more complex, ctDNA can also be secreted. You do not have to go deep, but please revise the sentence.

Sentence has been revised to acknowledge that the mechanism by which ctDNA appears in plasma is not fully understood. As suggested we have revised this sentence in the introduction to include necrosis.

3. The numbers in Figure 1 and Table 3 are not the same, seems like one recurrence is reported differently.

Apologies. This has been corrected.

4. Results: Why not also report specificity (96%) and sensitivity (40

We did mention this in the results in reference to the PPV and NPV, but we have now included the sensitivity and specificity in the results section and deleted the absolute values from the discussion.  Lines 155-158- read:

“As shown in Table 3, the rate of detectable ctDNA without disease progression or false positive results was low, with a specificity of 0.96. However, the rate of undetectable ctDNA in patients who developed disease progression or false negatives was much higher, resulting in a poor sensitivity of 0.33.”

Round 2

Reviewer 2 Report

The authors have answered to all my concerns.